# We Need to Talk About Reproducibility in NLP Model Comparison

**Yan Xue[1], Xuefei Cao[2], Xingli Yang[3], Yu Wang[4], Ruibo Wang[4], Jihong Li[4]**

1. School of Computer and Information Technology, Shanxi University, Taiyuan, 030006
2. School of Automation and Software Engineering, Shanxi University, Taiyuan, 030006
3. School of Mathematical Sciences, Shanxi University, Taiyuan, 030006
4. School of Modern Education Technology, Shanxi University, Taiyuan, 030006

202012407011@email.sxu.edu.cn
{caoxuefei, yangxingli, wangyu, wangruibo, lijh}@sxu.edu.cn

## Abstract

NLPers frequently face reproducibility crisis in a comparison of various models of a real-world NLP task. Many studies have empirically showed that the standard splits tend to produce low reproducible and unreliable conclusions, and they attempted to improve the splits by using more random repetitions. However, the improvement on the reproducibility in a comparison of NLP models is limited attributed to a lack of investigation on the relationship between the reproducibility and the estimator induced by a splitting strategy. In this paper, we formulate the reproducibility in a model comparison into a probabilistic function with regard to a conclusion. Furthermore, we theoretically illustrate that the reproducibility is qualitatively dominated by the signal-to-noise ratio (SNR) of a model performance estimator obtained on a corpus splitting strategy. Specifically, a higher value of the SNR of an estimator probably indicates a better reproducibility. On the basis of the theoretical motivations, we develop a novel mixture estimator of the performance of an NLP model with a regularized corpus splitting strategy based on a blocked $3 \times 2$ cross-validation. We conduct numerical experiments on multiple NLP tasks to show that the proposed estimator achieves a high SNR, and it substantially increases the reproducibility. Therefore, we recommend the NLP practitioners to use the proposed method to compare NLP models instead of the methods based on the widely-used standard splits and the random splits with multiple repetitions.

## 1 Introduction

In NLP domain, the reproducibility of empirical experimental conclusions has attracted most researchers' attention because the reproducibility crisis becomes more and more serious along with the increasing complexity of the proposed model architectures for a specific NLP task (Belz et al., 2021). In particular, in the past few decades, many researchers found that the majority of the conclusions in model comparisons can not be well reproduced in similar or slightly different experimental settings involving many real-world NLP tasks (Berg-Kirkpatrick et al., 2012; Søgaard et al., 2014; Belz et al., 2023), including POS tagging, parsing, text summarization, machine translation and so on.

Although a checklist methodology has been recommended in many NLP conferences, the checklist could merely address the reproducibility problems in the levels of methods and results rather than in the level of inferential conclusions (Bouthillier et al., 2019). Considering that about 70% of papers had an unpromising low reproducibility (Belz et al., 2021), whether a conclusion in an NLP model comparison is reproducible if one draw it from a different experimental setup is still an important problem that is not sufficiently addressed. In actual, the reproducibility of a conclusion depends on multiple variates or factors in an NLP experiment, such as source code, text corpus and so on.

An important factor that affects the reproducibility is the splitting strategy of a text corpus, which has been well recognized in recent years. As a thumb rule, standard splits on a text corpus are frequently used to ensure a fair comparison. However, Gorman and Bedrick (2019) found that the conclusion obtained on the standard splits is less reproducible and instable, and thereby they recommended random splits for NLP practitioners. Different from their views, Søgaard et al. (2021) pointed out that the random splits lead to overly optimistic performance estimators and harm the NLP model comparison. Thus, they recommended multiple biased splits for assessing NLP models. Moreover, Rob (2021) proposed a novel splitting strategy and introduced a tune set to select an optimal neural model in hundreds of epochs for fairness in subsequent comparisons. These studies provided sufficient empirical evidences to show the importance of a splitting strategy on the reproducibility

in a comparison but lacking of a necessary theoretical investigation on the relationship between the reproducibility and the estimator induced from a splitting strategy. Thus, the improvement on the standard splits is limited. Essentially, various splitting strategies lead to different estimators of the performance of an NLP model. The variance and the expectation of a performance estimator are critical to the reproducibility of the comparison of two NLP models. Therefore, in order to select an optimal splitting strategy and a reasonable estimation method, it is necessary to establish the theoretical relationship between the reproducibility and a performance estimator in a model comparison.

In this paper, we formulate the reproducibility in an NLP model comparison from a probabilistic perspective and theoretically establish the relationship between the reproducibility and the signal-to-noise ratio (SNR) of an estimator of model performance. We illustrate that an estimator with a higher SNR probably leads to a better reproducibility in a comparison. Therefore, we establish an optimization problem to maximize the SNR of an estimator in a model comparison. We further summarize that the solution to the problem should satisfy two criteria with regard to the variance and the expectation of the estimator, respectively. Motivated by the criteria, we develop a novel method that uses a blocked $3 \times 2$ cross-validation (BCV) coupled with a mixture estimator of NLP model performance and provide theoretical explanations of the method. We conduct numerical experiments on multiple NLP tasks to show that the proposed method produces an estimator with a high SNR and substantially increases the reproducibility of a conclusion in an NLP model comparison.

## 2 Reproducibility in Model Comparison

In this paper, we merely consider the reproducibility in a comparison of two NLP models from the perspectives of splitting strategy and estimation method. When the number of models is larger than two, the comparison can be decomposed into a collection of comparisons of any two NLP models among the candidates, and the comparison results on multiple pairs of models can be further reduced toward the final result with a proper correction, such as the Bonferroni procedure (Dror et al., 2017).

We denote a text corpus as $D_n$ where $n$ is the corpus size, and all the samples in $D_n$ are IID-sampled

from population $\mathfrak{D}$. Let $\mathcal{A}$ and $\mathcal{B}$ be two NLP models built on the corpus $D_n$, and the corresponding model instances are denoted as $\mathcal{A}(D_n)$ and $\mathcal{B}(D_n)$. Furthermore, let $\nu$ be a commonly-used performance metric, such as accuracy, $F_1$ score and so on. Without loss of generality, we assume that higher values of $\nu$ indicate better performance. Furthermore, if a previous empirical study compared the performance of models $\mathcal{A}(D_n)$ and $\mathcal{B}(D_n)$ based on a particular splitting strategy $\mathbb{S}$ and published a conclusion of $\nu(\mathcal{A}(D_n)) > \nu(\mathcal{B}(D_n))$ in a paper, then we have to ask a question: *How the splitting strategy $\mathbb{S}$ affects the reproducibility of the conclusion $\nu(\mathcal{A}(D_n)) > \nu(\mathcal{B}(D_n))$?* For example, assume the conclusion is obtained based on standard splits. If another research group uses slightly different splits to perform a similar comparison, to what extent the conclusion is reproduced?

Formally, conditioned on a published model comparison conclusion $\nu(\mathcal{A}(D_n)) > \nu(\mathcal{B}(D_n))$, the reproducibility in a comparison can be formalized into the following probabilistic form.

$$P(\hat{\nu}_{\mathbb{S}}(\mathcal{A}(D_n)) > \hat{\nu}_{\mathbb{S}}(\mathcal{B}(D_n))) = P(\hat{\nu}_{\mathbb{S}}^{\mathcal{A}-\mathcal{B}} > 0), \quad (1)$$

where $\hat{\nu}_{\mathbb{S}}^{\mathcal{A}-\mathcal{B}}$ denotes the estimator of the difference between the performance of models $\mathcal{A}(D_n)$ and $\mathcal{B}(D_n)$ based on randomly-generated splits in $\mathbb{S}$, and the randomness in Eq. (1) comes from $\mathbb{S}$.

An optimal splitting strategy $\mathbb{S}$ should make the reproducibility in a comparison as large as possible. Thus, we can purse the optimal splitting strategy and estimator by solving the following optimization problem.

$$(\mathbb{S}^*, \hat{\nu}^*) = argmax_{\mathbb{S}, \hat{\nu}} P(\hat{\nu}_{\mathbb{S}}^{\mathcal{A}-\mathcal{B}} > 0), \quad (2)$$

where the splitting strategy and estimator should be optimized simultaneously because different estimators can be constructed in an identical splitting strategy.

It is hard to solve the optimization problem in Eq. (2) since the distribution of $\hat{\nu}_{\mathbb{S}}^{\mathcal{A}-\mathcal{B}}$ is unknown and varies over different NLP corpora. Therefore, to develop an appropriate surrogation of Eq. (2), conditioned on the conclusion of $\nu(\mathcal{A}(D_n)) > \nu(\mathcal{B}(D_n))$, we apply the one-side Chebyshev inequality on Eq. (1) and obtain the following lower bound.

$$P(\hat{\nu}_{\mathbb{S}}^{\mathcal{A}-\mathcal{B}} > 0) \geq \frac{\text{SNR}^2(\hat{\nu}_{\mathbb{S}}^{\mathcal{A}-\mathcal{B}})}{1 + \text{SNR}^2(\hat{\nu}_{\mathbb{S}}^{\mathcal{A}-\mathcal{B}})}, \quad (3)$$

where $\text{SNR}(\hat{\nu}_\mathbb{S}^{\mathcal{A}-\mathcal{B}})$ is the SNR of the estimator $\hat{\nu}_\mathbb{S}^{\mathcal{A}-\mathcal{B}}$ with the following form.

$$\text{SNR}(\hat{\nu}_\mathbb{S}^{\mathcal{A}-\mathcal{B}}) = \frac{E[\hat{\nu}_\mathbb{S}^{\mathcal{A}-\mathcal{B}}]}{\sqrt{Var[\hat{\nu}_\mathbb{S}^{\mathcal{A}-\mathcal{B}}]}}, \qquad (4)$$

where $E[\cdot]$ and $Var[\cdot]$ stand for the expectation and variance of an estimator, respectively.

Eq. (3) provides a lower bound of the reproducibility. The lower bound is a monotonically increasing function with regard to the SNR. The lower bound further illustrates that a small variance and a slightly large expectation of $\hat{\nu}_\mathbb{S}^{\mathcal{A}-\mathcal{B}}$ lead to a high value of the SNR, which probably indicates a better reproducibility. Therefore, instead of directly addressing the problem in Eq. (2), we aim to figure out a proper solution to the next best optimization problem as follows.

$$(\mathbb{S}^*, \hat{\nu}^*) = argmax_{\mathbb{S}, \hat{\nu}} \text{SNR}(\hat{\nu}_\mathbb{S}^{\mathcal{A}-\mathcal{B}}). \qquad (5)$$

Furthermore, according to the definition of SNR in Eq. (4), the solution to Eq. (5) should satisfy the following two criteria.

**Criterion I.** The optimal splitting strategy and estimation method should reduce the variance $Var[\hat{\nu}_\mathbb{S}^{\mathcal{A}-\mathcal{B}}]$ as much as possible.

**Criterion II.** The optimal splitting strategy and estimation method should maximize the expectation $E[\hat{\nu}_\mathbb{S}^{\mathcal{A}-\mathcal{B}}]$ as much as possible.

## 3 Our Approach

Although the optimization problem in Eq. (2) is relaxed into a second best problem in Eq. (5), the latter is still not solved through an analytic methodology. Despite this, the two criteria obtained from Eq. (5) may shed light on a heuristic design of a better splitting strategy and estimation method.

### 3.1 $3 \times 2$ **BCV**

According to **Criterion I**, a reasonable splitting strategy to reduce the variance of $\hat{\nu}_\mathbb{S}^{\mathcal{A}-\mathcal{B}}$ is using more splits in $\mathbb{S}$, which has been empirically verified by many researchers (Moss et al., 2018; Gorman and Bedrick, 2019). Nevertheless, the variance depends not only on the number of splits but also on its splitting ratio. Specifically, if a large portion of $D_n$ is used in training, then any two training sets in two splits of $\mathbb{S}$ possess a large overlapping samples that introduces unnecessary correlations

| Repetition Index | Fold 1 | Fold 2 |
|:---:|:---:|:---:|
| 1 | $B_{(1)}, B_{(2)}$ | $B_{(3)}, B_{(4)}$ |
| 2 | $B_{(2)}, B_{(4)}$ | $B_{(1)}, B_{(3)}$ |
| 3 | $B_{(1)}, B_{(4)}$ | $B_{(2)}, B_{(3)}$ |

Table 1: Splitting rules in a $3 \times 2$ BCV.

in $\hat{\nu}_\mathbb{S}^{\mathcal{A}-\mathcal{B}}$ and enlarges the variance. As illustrated in the plots in the first row of Figure 2, when a splitting ratio of 8:1 is used, the variance can not be efficiently reduced even using more splits. On the basis of a similar observation, Dietterich (1998) decided to use a splitting ratio of 5:5 and recommended a usage of repeated two-fold CV in a model comparison because the expectation of the number of overlapping samples between any two two-fold CVs is only $n/4$ (Markatou et al., 2005).

The effectiveness of repeated two-fold CV on a model comparison has been showed in a series of studies, including but not limited to the work of (Alpaydin, 1999; Yildiz, 2013; Wang et al., 2014; Wang and Li, 2019). Moreover, a novel version of the repeated two-fold CV, named $m \times 2$ BCV, that leads to a smaller variance in $\hat{\nu}_\mathbb{S}^{\mathcal{A}-\mathcal{B}}$ (Wang et al., 2017a), achieves a better performance in model comparison (Wang et al., 2017b).

A $3 \times 2$ BCV, recommended in this paper, is a specific version of $m \times 2$ BCV with $m = 3$. The construction of a $3 \times 2$ BCV is straightforward: divide a corpus $D_n$ into four equal-sized blocks, namely $B_{(1)}, B_{(2)}, B_{(3)}, B_{(4)}$, and combine the blocks to form the splits according to the rules in Table 1.

A $3 \times 2$ BCV possesses at least four advantages for ensuring the reproducibility in a comparison of NLP models: (1) a $3 \times 2$ BCV leads to a smaller variance of an estimator than a randomly generated splits with a size of 6 and a ratio of 5:5 (Wang et al., 2019); (2) a $3 \times 2$ BCV is straightforward to use thanks to its simple combination rules in Table 1; (3) when the sizes of a training set and a validation set is equal, certain frequency distributions over linguistic units of the training and validation sets are consistent with a relatively high probability (Wang and Li, 2019); and (4) each sample in $D_n$ occurs with the same counts in all training sets of a $3 \times 2$ BCV, which facilitates us to design an aggregated estimator based on majority voting and is firstly proposed in this paper.

| Validated Block | Basis Models | Basis Predictions | Vote Predictions |
|:---:|:---:|:---:|:---:|
| $B_{(1)}$ | $\mathcal{A}_{(2)(3)}, \mathcal{A}_{(2)(4)}, \mathcal{A}_{(3)(4)}$ | $\{\hat{y}_{(2)(3),(1)}\}, \{\hat{y}_{(2)(4),(1)}\}, \{\hat{y}_{(3)(4),(1)}\}$ | $\{\hat{y}_{(1)}^{v}\}$ |
| $B_{(2)}$ | $\mathcal{A}_{(1)(3)}, \mathcal{A}_{(1)(4)}, \mathcal{A}_{(3)(4)}$ | $\{\hat{y}_{(1)(3),(2)}\}, \{\hat{y}_{(1)(4),(2)}\}, \{\hat{y}_{(3)(4),(2)}\}$ | $\{\hat{y}_{(2)}^{v}\}$ |
| $B_{(3)}$ | $\mathcal{A}_{(1)(2)}, \mathcal{A}_{(1)(4)}, \mathcal{A}_{(2)(4)}$ | $\{\hat{y}_{(1)(2),(3)}\}, \{\hat{y}_{(1)(4),(3)}\}, \{\hat{y}_{(2)(4),(3)}\}$ | $\{\hat{y}_{(3)}^{v}\}$ |
| $B_{(4)}$ | $\mathcal{A}_{(1)(2)}, \mathcal{A}_{(1)(3)}, \mathcal{A}_{(2)(3)}$ | $\{\hat{y}_{(1)(2),(4)}\}, \{\hat{y}_{(1)(3),(4)}\}, \{\hat{y}_{(2)(3),(4)}\}$ | $\{\hat{y}_{(4)}^{v}\}$ |

Table 2: Aggregation rules in a vote estimation on $3 \times 2$ BCV.

### 3.2 Mixture Estimation for $\hat{\nu}_{\mathbb{S}}^{\mathcal{A}-\mathcal{B}}$

In **Criterion II**, the expectation $E[\hat{\nu}_{\mathbb{S}}^{\mathcal{A}-\mathcal{B}}]$ means the expectation of the difference of the estimators in two models $\mathcal{A}(D_n)$ and $\mathcal{B}(D_n)$. In theory, the expectation in a single model is frequently regarded as monotonically increasing with regard to a corpus size of $n$. However, for the expectation of the difference, i.e., $E[\hat{\nu}_{\mathbb{S}}^{\mathcal{A}-\mathcal{B}}]$, it is not reasonable to assume that it is increasing with regard to $n$. In order to express clearly, we abbreviate $E[\hat{\nu}_{\mathbb{S}}^{\mathcal{A}-\mathcal{B}}]$ as $\nu_n$ of which the subscript is the size of a training set. Furthermore, we consider the following two cases about $\nu_n$.

**Case I.** $\nu_n$ is an increasing function with regard to $n$. It shows that as $n$ increases, model $\mathcal{A}(D_n)$ is substantially better than $\mathcal{B}(D_n)$.

**Case II.** $\nu_n$ is decreasing with regard to $n$. It means that when $n$ becomes large, the performance superiority of model $\mathcal{A}(D_n)$ is gradually disappearing.

Although the relationship between $\nu_n$ and $n$ in a real-world NLP task may be more complex, the assumptions in the above-mentioned two cases are more loose than that in previous studies which frequently assumed that the value of $\nu_n$ is almost unchanged with regard to $n$ (Dietterich, 1998; Nadeau and Bengio, 2003; Wang and Li, 2019). Moreover, we assume that $sgn(\nu_{n/2}) = sgn(\nu_n)$ where $sgn(\cdot)$ is the sign function. We consider that the assumption is natural when $n$ is slightly large.

We adopt two different estimation methods for the two cases respectively. In actual, in a $3 \times 2$ BCV, researchers frequently obtain six hold-out estimators of $\nu$. The six estimators correspond to the six folds in Table 1. In theory, each hold-out estimator is unbiased to $\nu_{n/2}$.

For **Case I**, because $\nu_n$ is increasing with regard to $n$, we obtain that $\nu_{n/2} \leq \nu_n$. Therefore, the hold-out estimators and the average of the estimators is an unpromising estimation method that can

not satisfy **Criterion II** in Section 3.2. In contrast, an aggregation estimation method based on majority voting leads to a promising estimator of which the expectation is more closer to $\nu_n$ than that of the hold-out estimators (Yang et al., 2023). We abbreviate the aggregation estimation as vote estimator and denote it as $\hat{\nu}_{vote}^{\mathcal{A}-\mathcal{B}}$.

On the basis of the theoretical results in (Yang et al., 2023), we obtain that the vote estimator satisfies the following two properties.

$$\nu_{n/2} \leq E[\hat{\nu}_{vote}^{\mathcal{A}-\mathcal{B}}]. \tag{6}$$

$$|E[\hat{\nu}_{vote}^{\mathcal{A}-\mathcal{B}}] - \nu_n| \leq |\nu_n - \nu_{n/2}|. \tag{7}$$

Thus, the vote estimator is more suitable to **Criterion II** than the hold-out estimators and their average.

We demonstrate the computation of a vote estimator $\hat{\nu}_{vote}^{\mathcal{A}-\mathcal{B}}$ in a supervised NLP model. Denote a supervised text corpus as $D_n = \{(\mathbf{x}_j, y_j)\}_{j=1}^{n}$ where $\mathbf{x}_j$ is the input text and $y_j$ corresponds to the supervised labels. For example, for a semantic role labeling task, $\mathbf{x}$ is a sentence with a pre-defined target word, and $y$ is the gold IOB2 labels for representing the semantic role chunks in the sentence. On the basis of a $3 \times 2$ BCV, six basis models are obtained on $D_n$, denoted as $\mathcal{A}_{(1)(2)}, \mathcal{A}_{(3)(4)}, \mathcal{A}_{(2)(4)}, \mathcal{A}_{(1)(3)}, \mathcal{A}_{(1)(4)}, \mathcal{A}_{(2)(3)}$, of which the subscripts correspond to the indices of the blocks in Table 1. Then, we use three of the basis models to validate the samples in each block and obtain three sets of basis predictions. Furthermore, on the basis of majority voting, we aggregate the three sets into a set of vote predictions. The aggregation rules are given in Table 2. Finally, we compare the vote predictions $\{\hat{y}_j^v\}_{j=1}^{n} = \{\hat{y}_{(1)}^{v}\} \cup \{\hat{y}_{(2)}^{v}\} \cup \{\hat{y}_{(3)}^{v}\} \cup \{\hat{y}_{(4)}^{v}\}$ and the gold predictions $\{y_j\}_{j=1}^{n}$ to obtain the vote estimator of model $\mathcal{A}$, denoted as $\hat{\nu}_{vote}^{\mathcal{A}}$. In a similar way, we can compute the vote estimator $\hat{\nu}_{vote}^{\mathcal{B}}$ for model $\mathcal{B}$. Then, the vote estimator for **Case I** is $\hat{\nu}_{vote}^{\mathcal{A}-\mathcal{B}} = \hat{\nu}_{vote}^{\mathcal{A}} - \hat{\nu}_{vote}^{\mathcal{B}}$.

For **Case II**, because $\nu_n$ is decreasing with $n$, we obtain that $\nu_{n/2} \geq \nu_n$. Thus, we can prefer the average of the six hold-out estimators because the averaged estimator is unbiased to $\nu_{n/2}$, and the averaged estimator is more stable and more suitable to **Criteria I** and **II** than the hold-out estimators and the vote estimator. We denote the averaged estimator as $\hat{\nu}_{avg}^{\mathcal{A}-\mathcal{B}}$.

Unfortunately, for a real-world NLP model, whether its $\nu_n$ satisfies **Case I** or **Case II** is unknown. Thus, we have to construct a novel mixture estimator to comprehensively consider the two cases. The mixture estimator has a form as follows.

$$\hat{\nu}_{mix}^{\mathcal{A}-\mathcal{B}} = \Delta \cdot \hat{\nu}_{vote}^{\mathcal{A}-\mathcal{B}} + (1 - \Delta) \cdot \hat{\nu}_{avg}^{\mathcal{A}-\mathcal{B}}, \quad (8)$$

where $\Delta$ is an indicator function such that $\Delta = 1$ when $\hat{\nu}_{vote}^{\mathcal{A}-\mathcal{B}} > \hat{\nu}_{avg}^{\mathcal{A}-\mathcal{B}}$ and $\Delta = 0$ otherwise. In a comparison, we consider that when $\Delta = 1$, **Case I** has a higher probability to occur than **Case II**, and vice versa. Therefore, we use $\Delta$ as a switcher to adaptively select an estimator between $\hat{\nu}_{vote}^{\mathcal{A}-\mathcal{B}}$ and $\hat{\nu}_{avg}^{\mathcal{A}-\mathcal{B}}$.

### 3.3 Baselines

Four baselines are considered in this paper.

**Standard splits (ST).** The conventional standard splits on a corpus is static and designed in advance. The standard splits frequently use a ration of 8:1 to split a text corpus into a training set and a validation set (Gorman and Bedrick, 2019). The static property in standard splits tend to produce an obvious over-fitting phenomenon attributing to the well-known "file-drawer" problem (Scargle, 2000) and the adaptive usage problem of a fixed hold-out (Dwork et al., 2015). Therefore, to eliminate the static property, we randomly generate splits with a ratio of 8:1 in a hold-out manner and use them as our first baseline. We also named them as standard splits.

**Random splits (RS).** The random splits for NLP model comparison are recommended in (Gorman and Bedrick, 2019). They proposed to use multiple repetitions of hold-out validation with a splitting ratio of 8:1 to compare NLP models. The repetition count is set to an integer in [3, 10] in this paper, which is the same with (Søgaard et al., 2021). It is noted that a vote estimator is hard to construct in random splits because the occurrence counts of the samples in all training sets of random splits are different. Moreover, for a fair comparison, we mainly compare our $3 \times 2$ BCV with the random splits with 6 repetitions.

$3 \times 2$ **BCV with an averaged estimator (Avg).** We adopt the averaged estimator $\hat{\nu}_{avg}^{\mathcal{A}-\mathcal{B}}$ based on a $3 \times 2$ BCV as a baseline in all experiments. The $3 \times 2$ BCV averaged estimator has been studied in a comparison task of supervised classifiers (Wang et al., 2015).

$3 \times 2$ **BCV with a vote estimator (Vote).** The vote estimator $\hat{\nu}_{vote}^{\mathcal{A}-\mathcal{B}}$ based on a $3 \times 2$ BCV is used as a baseline over all NLP model comparisons. This baseline is first developed in this paper.

Moreover, the training and validation sets produced by the splitting strategy proposed in (Søgaard et al., 2021) possess biased and different distributions. This characteristic differs from the fundamental assumption in our paper that the training and validation sets should be drawn from an identical distribution $\mathfrak{D}$. Therefore, we don't consider the method in (Søgaard et al., 2021) as our baseline. The method in (Rob, 2021) focuses on a situation of tuning a neural NLP model that is out of our consideration. Thus, we don't use the method in (Rob, 2021) as a baseline.

We named our proposed method, i.e., the mixture estimator based on a $3 \times 2$ BCV, as **Mixture**.

## 4 Experiments

We compare our proposed method with the baselines in Section 3.3 on the following NLP tasks.

**Semantic role labeling task (SRL)**: Identify the boundaries of all semantic role chunks in a Chinese sentence with gold word segmentation and a pre-defined target word based on the Chinese FrameNet annotation convention. A Chinese FrameNet training corpus is used as $D_n$ that contains 35,473 Chinese sentences and 78,749 semantic chunks. A Bi-LSTM algorithm coupled with linguistic features of word, POS, target word, and frame name, is used to train and validate an SRL model. Considering the embedding size of a target word is an important hyper-parameter, we compare the two SRL models obtained from the settings of the embedding size being 10 and 20, respectively.

**Named entity recognition task (NER)**: Identify the boundaries of all NER chunks without recognizing their type labels. We use CoNLL 2003 English NER training set as $D_n$ that contains 14,987 sentences and 23,499 chunks. The corpus contains four types of named entities, including PER, LOC, ORG, and MISC. We consider the NER task as a sequence labeling problem and use CRFs to train and validate an NER model. Two labeling sets of

| Task | Sample Means (%) | | Standard Errors ($\times 10^{-4}$) | | Confidence Intervals (%) | |
|---|---|---|---|---|---|---|
| | $\mathcal{A}(D_n)$ | $\mathcal{B}(D_n)$ | $\mathcal{A}(D_n)$ | $\mathcal{B}(D_n)$ | $\mathcal{A}(D_n)$ | $\mathcal{B}(D_n)$ |
| SRL | 61.98 | 61.60 | 5.90 | 5.94 | [61.87, 62.10] | [61.48, 61.71] |
| NER | 89.38 | 89.20 | 1.86 | 2.07 | [89.34, 89.42] | [89.16, 89.24] |
| Chunking | 82.25 | 82.20 | 1.16 | 1.10 | [82.23, 82.27] | [82.18, 82.22] |

Table 3: Confidence intervals of model performance on the three NLP tasks.

IOB2 and IOBES are used to induce two different NER models that are compared in our experiments.

**Chinese base-phrases chunking task (Chunking)**: Identify the boundaries of all base-phrases in a Chinese sentence with gold word segmentation. The Chinese base-phrases corpus is developed by Zhang and Zhou (2002). The corpus is used as $D_n$ that contains 14,248 sentences and 186,319 chunks. We use CRFs based on different labeling sets of IOB2 and IOBES to obtain two different chunking models that are compared in our experiments.

The above three tasks usually adopt precision, recall, and $F_1$ score as evaluation metrics. Due to space limitation, we merely report the results about $F_1$ score.

For the three NLP tasks, we consider the following empirical conclusions in model comparisons.

- For the SRL task, we consider that the embedding size of 20 (model $\mathcal{A}$) is better than that of 10 (model $\mathcal{B}$) as a conclusion.

- For the NER task, IOBES (model $\mathcal{A}$) is empirically better than IOB2 (model $\mathcal{B}$).

- For the chunking task, the conclusion of IOB2 (model $\mathcal{A}$) outperforming IOBES (model $\mathcal{B}$) is considered because a base-phrase chunk is frequently short, and IOBES is too delicate.

To check the correctness of the above empirical conclusions, i.e., $\nu(\mathcal{A}(D_n)) > \nu(\mathcal{B}(D_n))$, we conduct numerical simulations to obtain the performance of models $\mathcal{A}$ and $\mathcal{B}$. Specifically, we combine the original training and test sets and randomly sample a training set and a test set in a without-replacement manner from the combined data set. The sampling ratio is the same as the original splitting ratio. Then, we independently perform the process 1,000 times and use the sample mean of the 1,000 values of estimators $\hat{\nu}(\mathcal{A}(D_n))$ and $\hat{\nu}(\mathcal{B}(D_n))$ to approximate $\nu(\mathcal{A}(D_n))$ and $\nu(\mathcal{B}(D_n))$. Considering that the 1,000 estimators are IID, we further compute the

standard errors and the confidence intervals with $\alpha = 0.05$. The results listed in Table 3 illustrate that for each of the three NLP tasks, the confidence intervals between models $\mathcal{A}(D_n)$ and $\mathcal{B}(D_n)$ are not overlapped. Therefore, the empirical conclusions are considered to be correct.

On the basis of a text corpus and a splitting strategy, we randomly generate the corresponding splits and use the splits to train and validate the compared NLP models and obtain the corresponding estimators of the difference of the performance of the two compared models. This process is independently repeated in 1,000 times, and we obtain 1,000 values of an estimator that is used to compute the corresponding SNR and reproducibility.

For the computation of the SNR, we approximate the expectation and the variance of an estimator with the sample mean and the sample variance of the 1,000 values of an estimator. Then, we use Eq. (4) to compute the numerical value of the SNR.

For the computation of the reproducibility, we compute the proportion of the 1,000 values greater than zero. Then, we obtain the numerical value of the reproducibility by replacing the probability in Eq. (1) with the empirical proportion.

We organize experiments into three groups to illustrate (1) the relationship between the SNR and the reproducibility; and (2) the distribution of an estimator in each of different methods; and (3) the reproducibility over different baselines and our method.

## 4.1 Experiments on SNR vs. Reproducibility

We first perform experiments to illustrate the relationship between the SNR and the reproducibility based on random splits. Specifically, we vary the repetition count of random splits from 3 to 10 and depict the corresponding SNR and reproducibility in Figure 1.

Intuitively, when the repetition count increases, the SNR becomes large. From Figure 1, it is obtained that with an increasing SNR, the re-

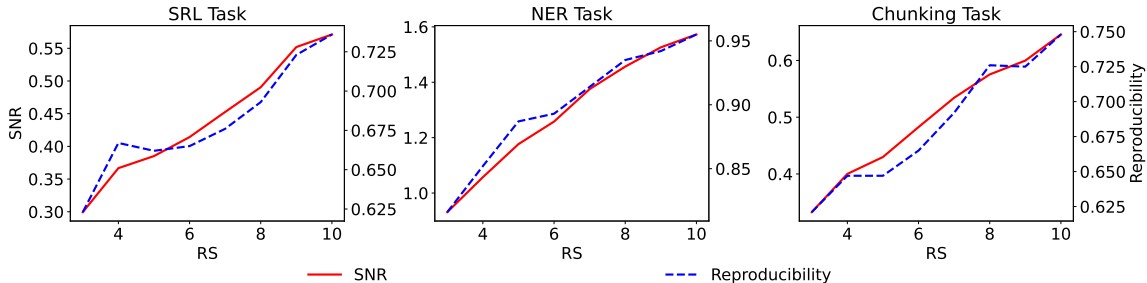

Figure 1: Illustration of the relationship between reproducibility and SNR.

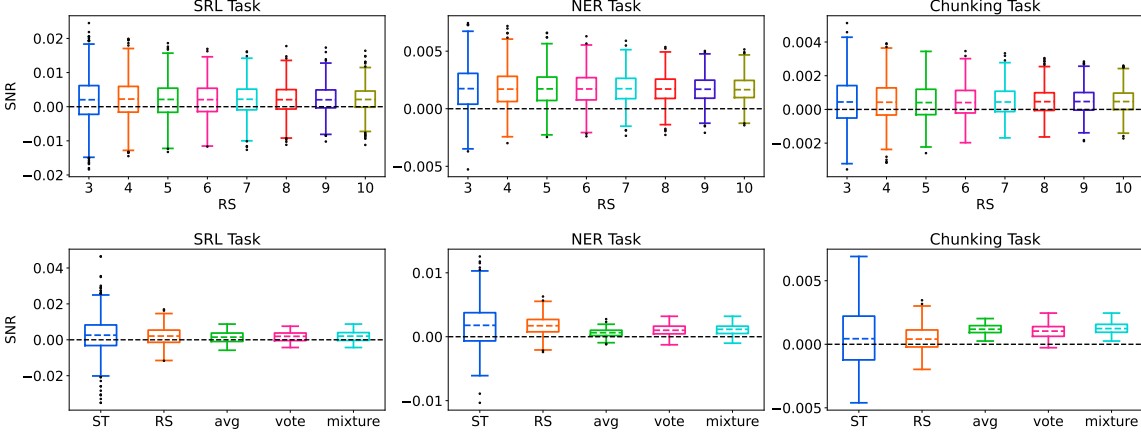

Figure 2: Box plots of the distributions of the estimators in different methods.

producibility increases qualitatively. The increase doesn't strictly hold in a quantitative manner because the analytic form in the right-hand side of Eq. (3) with regard to the SNR is a lower bound rather than an equivalent form of the reproducibility. As showed in the third plot in Figure 1, when the repetition count changes from 4 to 5, the SNR increases but the reproducibility slightly decreases. A few similar observations can also be found in Table 4. Despite this, the qualitative relationship still allow us to maximize the SNR as the second best choice to find a better splitting strategy and estimation method for ensuring a high reproducibility.

### 4.2 Experiments on Distribution of $\hat{\nu}_{\mathbb{S}}^{\mathcal{A}-\mathcal{B}}$

We further provide the distributions of the values of the estimators in different baselines and our method in box-plots showed in Figure 2. The horizontal dotted line indicates a zero value.

The three plots in the first row of Figure 2 provide the distributions of the estimators in random splits over the three NLP tasks with an increasing repetition count. From the three plots, we obtain that when the repetition count is greater than 5, the width of the box-plot keeps almost unchanged. In other words, the variance and the SNR of the

estimator in random splits cannot be efficiently reduced even using more splits. An important reason is that the overlapping samples between any two training sets in random splits account for a large proportion (about 70%) of $D_n$.

In the second row of Figure 2, the distributions of the estimators in different methods are compared.

From the perspective of **Criterion I** in Section 2, a common observation can be found over three NLP tasks. Standard splits and random splits possess a larger variance than the other three methods based on a $3 \times 2$ BCV that verifies the minimal variance property of an estimator in a $3 \times 2$ BCV, as described in Section 3.1.

Moreover, with the perspective of **Criterion II**, we concentrate on the expectations of the estimators. Several interesting observations are found over the three NLP tasks.

For the SRL task, considering the plot in the left bottom of Figure 2, we find that the expectations of the estimators in different methods are almost identical, i.e., $\nu_{n/2} \approx \nu_n$. That corresponds to the assumption adopted in previous studies. In this case, the estimator that owns the smallest variance performs better in the model comparison.

| Task | Splitting Strategy | ST
8:1 hold-out | RS
6 repetitions of 8:1 | Avg | Vote | Mixture |
|------|--------------------|--------------------|---------------------------|-----|------|---------|
| | | | | \multicolumn{3}{c}{$3 \times 2$ BCV} | | |
| SRL | SNR | 0.181 | 0.414 | 0.465 | 0.645 | **0.684** |
| | Reproducibility | 0.636 | 0.665 | 0.659 | 0.694 | **0.714** |
| NER | SNR | 0.510 | 1.258 | 0.889 | 1.179 | **1.421** |
| | Reproducibility | 0.678 | 0.893 | 0.830 | 0.900 | **0.940** |
| Chunking | SNR | 0.195 | 0.483 | **3.200** | 1.818 | 3.063 |
| | Reproducibility | 0.590 | 0.665 | **1.000** | 0.960 | **1.000** |

Table 4: SNR and reproducibility of different methods on the three NLP tasks.

For the NER task, the estimator in random splits has a higher expectation than the averaged estimator in a $3 \times 2$ BCV. The phenomenon indicates that **Case I** in Section 3.2 occurs, i.e., $\nu_{n/2} \leq \nu_n$. In this case, even the averaged estimator has the smallest variance, its SNR is not high (as shown in Table 4), because the expectation of the averaged estimator is small.

For the chunking task, the plot in the right bottom of Figure 2 shows that the expectation of random splits is lower than that of the averaged estimator. In other words, **Case II** in Section 3.2 holds, i.e., $\nu_{n/2} \geq \nu_n$, and the averaged estimator has a higher SNR.

From the above observations with regard to the expectation, we have to re-state that the relationship between $\nu_n$ and $n$ varies over different NLP tasks and is unknown in advance. This is an important motivation for us to develop a mixture estimation based on a $3 \times 2$ BCV.

Moreover, it is noted that we concentrate on the experimental settings of model $\mathcal{A}$ being slightly better than model $\mathcal{B}$ in Figure 2 because when model $\mathcal{A}$ is substantially better than model $\mathcal{B}$ (such as BERT vs. CRFs), all splitting strategies, including ST, RS, and $3 \times 2$ BCV, tend to achieve the best reproducibility (=1.0). Thus, it is hard to observe the difference between the effects of the different splitting strategies on the reproducibility even though our proposed method still owns a higher SNR.

### 4.3 Experiments on Reproducibility

Several experiments are performed to compute the reproducibility of different splitting strategies and estimation methods over the three NLP tasks. The results are given in Table 4. Several observations are obtained on Table 4.

The standard splits possess the lowest SNR and reproducibility that indicates the optimality of the selected NLP models based on standard splits cannot be guaranteed. This conclusion is similar with that in (Gorman and Bedrick, 2019; Rob, 2021).

The random splits have a higher SNR and reproducibility than the standard splits because the former uses more repetitions to make the estimator more stable. Thus, we consider the improvements in (Gorman and Bedrick, 2019) are effective and significant. However, the room for improvement on the reproducibility in random splits by using more splits is limited attributing to the large portion of the overlapping samples between the training sets.

Compared with the standard splits and the random splits, the estimators in a $3 \times 2$ BCV achieve a better SNR and reproducibility. Therefore, a $3 \times 2$ BCV is a better splitting strategy in an NLP model comparison. Furthermore, comparing the three estimators in a $3 \times 2$ BCV, we obtain that in most situations, the mixture estimator achieves the highest SNR and reproducibility. Thus, the mixture estimator in a $3 \times 2$ BCV, proposed in this paper, should be preferred in an NLP model comparison.

In summary, we recommend three guidelines:

(1) Standard splits should be used with caution in a comparison of two NLP models because standard splits may lead to a low reproducibility and unreliable conclusion in the comparison.

(2) A $3 \times 2$ BCV could be preferred when assessing and comparing NLP models. The averaged estimator and vote estimator on a $3 \times 2$ BCV can be considered because they achieve a promising reproducibility in the task of model comparison.

(3) NLP practitioners could give priority to the mixture estimator coupled with a $3 \times 2$ BCV in a comparison of two NLP models.

## 5 Related Work

In recent decades, a task of reproducing and/or replicating results and conclusions in published NLP papers has attracted many researchers' attention. Many studies have illustrated that the reproducibility in NLP models has a close relationship with many factors, including availability of source code (Arvan et al., 2022), randomization manner of instances in a text corpus (Bestgen, 2020), nature of an NLP task and under-specified or missing details in a published paper (Rim et al., 2020), readability of a published paper and number of hyperlinks (Akella et al., 2022), splitting strategy of a corpus (Gorman and Bedrick, 2019), statistical test used in an NLP task (Søgaard, 2013; Søgaard et al., 2014), tradition of publishing reproduction attempts (Cohen et al., 2016), and so on. Therefore, several systematic reviews and meta analysis related to the reproducibility in NLP models are conducted in recent years, including but not limited to (Cohen et al., 2018; Arvan et al., 2022; Belz et al., 2021, 2022, 2023). Furthermore, many effective methods are recommended to ensure a high reproducibility in an NLP study. For example, Dror et al. (2017) proposed a novel replicability analysis framework based on partial conjunction testing. Dror et al. (2019) developed a deep dominance method to properly compare neural NLP models. Belz (2022) recommended a quantitative measure of reproducibility from a metrological perspective. Magnusson et al. (2023) investigated the commonly-used checklist method.

Besides the above-mentioned studies, an attractive direction is to investigate the relationship between a splitting strategy of a text corpus and the reproducibility in a comparison of two NLP models. Moss et al. (2018) showed that the optimized values of hyper-parameters in an NLP model are highly sensitive to how a text data set is partitioned. Gorman and Bedrick (2019) illustrated that the popular standard splits are insufficient to reproduce a conclusion in the task of NLP model comparison. They recommended multiple random splits of a text corpus. Søgaard et al. (2021) further made an improvement on the splitting strategy of a text corpus and proposed biased splits for NLP practitioners. Rob (2021) introduced a tune set to make the optimization process of a neural NLP model more reliable. These studies shed lights on the direction and inspired us to further optimize the splitting strategy from a different perspective.

## 6 Conclusions

In this paper, we theoretically analyze the relationship between the reproducibility in an NLP model comparison and the SNR of an estimator of the difference of the performance of two NLP models. We further show that a higher value of the SNR probably indicates a better reproducibility. To increase the reproducibility, we formulate an optimization problem to maximize the SNR and further establish two criteria for figuring out an optimal splitting strategy and a better estimation method. We develop a novel method that uses a $3 \times 2$ BCV coupled with a mixture estimator and illustrate the superiority of the proposed method through multiple real-world NLP tasks. We recommend the NLP practitioners to use a $3 \times 2$ BCV and the mixture estimator to compare their NLP models.

In future, we aim to investigate a comparison of multiple NLP models and try our best to figure out a better method to improve the reproducibility of the comparison based on the splitting strategy of an $m \times 2$ BCV.

## 7 Limitations

Despite the novelty and benefits of our method for reproducibility of a conclusion in an NLP model comparison, it does include some drawbacks. In this paper, the theoretical analysis and experimental results illustrate that our proposed method can substantially improve the reproducibility of a conclusion in a comparison of two NLP models. However, we have not obtain an equivalent form of the reproducibility with regard to the SNR that indicates there is still a room for improvement from a theoretical perspective. In addition, our proposed method can be improved through adopting an $m \times 2$ BCV instead of a $3 \times 2$ BCV and regularizing the divergence between the empirical distributions of the training and validation sets in an $m \times 2$ BCV. These would form our future directions.

## Acknowledgement

We thank the anonymous reviewers for their helpful and constructive comments. This paper is supported by the National Natural Science Foundation of China (62076156, 61806115) and the Scientific and Technological Innovation Programs of Higher Education Institutions in Shanxi (2021L004). The experiments are supported by High Performance Computing System of Shanxi University. The corresponding author is Jihong Li.

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
