# OpenReview forum: "We Need to Talk About Reproducibility in NLP Model Comparison"
_EMNLP/2023/Conference — EMNLP 2023 Main_

### Official Review · Reviewer_zPdM · 2023-08-02

**Soundness:** 4

**Excitement:**

4: Strong: This paper deepens the understanding of some phenomenon or lowers the barriers to an existing research direction.

**Paper Topic And Main Contributions:**

This paper proposes a novel method to split a dataset in order to improve the reproducibility of the results, i.e., the same conclusions are drawn from different splits. The paper compares different splitting strategies on 3 different benchmarks and evaluates how many times the conclusion of system A is better than system B occurs (reproducibility). As a conclusion, their proposed method outperforms the rest of the candidates.

**Reasons To Accept:**

* This paper addresses the important fact of reproducibility in NLP.
* Their proposed approach outperforms the rest of the candidates based on their evaluation criteria.

**Reasons To Reject:**

* I am a bit concerned about the evaluation. Typically, when you evaluate a model, you compare the learnt distribution against the test dataset which represents the true distribution. And then, see which models performed better. However, in this paper is the other way around, you assume that model A is better than B and optimize how to split the dataset in order to satisfy the hypothesis. I understand that, in fact, dataset splitting could be a relevant factor in the model performance. But keeping the splits static makes sure that both models are compared in the same conditions (train/test splits). Additionally, if we just consider the estimations of standard splits, despite the NER task, it is not clear to me whether model A is significantly better than B. I think a significant test is required here before further analysis is performed.

**Reproducibility:**

4: Could mostly reproduce the results, but there may be some variation because of sample variance or minor variations in their interpretation of the protocol or method.

**Reviewer Confidence:**

2: Willing to defend my evaluation, but it is fairly likely that I missed some details, didn't understand some central points, or can't be sure about the novelty of the work.

**Typos Grammar Style And Presentation Improvements:**

* {line 248-250} “3 x 2 BCV is easy-to-use and has a low computational cost in training that is important in the study of reproducibility” This is not true, the computational cost depends on both model and data, and in cases of large models and large datasets, spending 3 times more training time could be very expensive. I do not say it is not necessary but is not cheap.
* The Related Work section is merged in the Introduction, I would prefer to have it in another different section.

---

> ### Author Rebuttal · Authors · 2023-08-28
>
> Thanks for your comments. The evaluation method in our paper is different from the typical evaluation method in many empirical NLP studies because our aim in the submitted paper is not the same as the aims in typical NLP model evaluation and comparison tasks.
>
> Our aim is to investigate the relationship between different splitting strategies and the reproducibility of a published conclusion in a model comparison (as mentioned in line 143 in our paper). An effective and direct evaluation method for the relationship is to numerically compute the true values of the reproducibility over different splitting strategies based on computationally intensive statistical simulations. Consider that the reproducibility of a conclusion is formulated into the following conditional probability (as same as Eq. (1) in our paper).
> $$P(\hat{\nu}_S(A(D_n))> \hat{\nu}_S(B(D_n))|\nu_S(A(D_n))> \nu_S(B(D_n))). \tag{1}$$
>
> To compute the probability, we should introduce a conclusion in advance as the condition of the probability in our evaluation experiments and then perform statistical simulations. The computation frequently executes thousands of numerical simulations, and thus our evaluation is substantially different from the typical evaluation methods in the NLP model evaluation and comparison tasks.
>
> The simulation-based evaluation method in our paper is very similar to the evaluation method on the comparisons of type I and II errors and replicabilities of different significance tests in a comparison task of machine learning algorithms (including but not related to [1-4]). In these studies, because type I and II errors are probabilities conditioned on a hypothesis, their evaluation methods assumed that a hypothesis holds and numerically simulated the values of type I and II errors and optimized a significance test to satisfy the assumed hypothesis. In the machine learning domain, the simulation-based evaluation method in these studies helps the researchers find several useful and interesting results. For example, a K-fold cross-validated t-test tends to produce false positive conclusions in an algorithm comparison, and repeated two-fold cross-validation achieves a better performance in a t-test than a hold-out validation and a K-fold cross-validation in the task of algorithm comparison. In the NLP domain, the simulation-based evaluation method makes it possible to explore the theoretical properties of new NLP models and methods. As a typical example, our paper uses the simulation-based evaluation method to investigate the relationship between a splitting strategy and reproducibility.
>
> You pointed out that keeping the splits static could ensure that all comparisons are performed in the same conditions. Despite this, the static splits frequently make the sequentially reported results have a tendency to overfit the static test set, and thus the subsequently published conclusions are probably false positive. This disadvantage of static splits is well known as the ''file-drawer'' problem in scientific inference [6] and is investigated in adaptive data analysis [7]. In particular, static splits possibly make the conclusions less reproducible and unstable [8]. Therefore, we try our best in the paper to find another splitting strategy for highly reproducible conclusions.
>
> We thank you for your constructive suggestion for using a significance test. We consider that a significance test can produce the significance of a conclusion rather than the reproducibility of the conclusion. Therefore, we carefully think that it may not be very necessary to use a significance test before our simulation-based evaluation. Moreover, in the NLP domain, how to select a suitable significance test for an NLP task and a specific evaluation metric (such as $F_1$-score) is still an open challenge [9]. As an important future direction, we will carefully consider how to construct an appropriate significance test in a comparison of NLP models to reduce the false positive conclusions.
>
> Finally, thanks for your warm reminder about the presentation improvements. We will carefully revise our presentation in the paper and add a related work section in the published version.
>
> [1] Dietterich, T. G. (1998). Approximate statistical tests for comparing supervised classification learning algorithms. Neural Computation 10(7): 1895-1923.
>
> [2] Alpaydin, E. (1999). Combined $5\times2$ cv F-test for comparing supervised classification learning algorithms. Neural Computation 11(8): 1885-1892.
>
> [3] Nadeau, C. and Y. Bengio (2003). Inference for the generalization error. Machine Learning 52(3): 239-281.
>
> [4] Bouckaert, R. R. and E. Frank (2004). Evaluating the replicability of significance tests for comparing learning algorithms. Advances in knowledge discovery and data mining, Springer: 3-12.
>
> [5] Wang, Y., et al. (2014). Blocked 3x2 cross-validated t-test for comparing supervised classification learning algorithms. Neural Computation 26(1): 208-235.
>
> [6] Jeffrey D Scargle. 2000. Publication bias: The ''file- drawer'' problem in scientific inference. Journal of Scientific Exploration, 14(1):91–106.
>
> [7] Dwork, C., et al. (2015). The reusable holdout: Preserving validity in adaptive data analysis. Science 349(6248): 636-638.
>
> [8] Kyle Gorman and Steven Bedrick. 2019. We Need to Talk about Standard Splits. In ACL, pages 2786–2791.
>
> [9] Søgaard, A., et al. (2014). What's in a p-value in NLP? In the Eighteenth Conference on Computational Natural Language Learning, pages 1-10.

---

### Official Review · Reviewer_wxeB · 2023-08-02

**Soundness:** 4

**Excitement:**

4: Strong: This paper deepens the understanding of some phenomenon or lowers the barriers to an existing research direction.

**Paper Topic And Main Contributions:**

The authors address the issue of reproducibility of the results of any two models under different data splitting strategies. They relate the reproducibility estimate to that of a 'signal-to-noise' ratio. Following the definition of the aforementioned quantity, they argue for a blocked 3x2 cross validation splitting strategy for which, in addition to a standard average estimator, they introduce two more, a majority vote estimator and a mixture estimator combining the previous two.  As as application for their experiments they target 3 NLP tasks.

**Questions For The Authors:**

A.  Following your observations from equations (1) to (5), did you consider exploring other splitting strategies than 3x2 BCV?
B. How does 3x2 BCV compare to mx2 BCV, for various m, in terms of the observations laid out in this paper?

**Reasons To Accept:**

The authors nicely lay out the motivation for the 3x2 BCV over alternative splitting strategies such as standard or random splits. The results also seem to back up the theory.

**Reasons To Reject:**

Most of the observations regarding 3x2 BCV have been previously made known, as the authors acknowledge at the end of Section 3.1.

**Reproducibility:**

4: Could mostly reproduce the results, but there may be some variation because of sample variance or minor variations in their interpretation of the protocol or method.

**Reviewer Confidence:**

3: Pretty sure, but there's a chance I missed something. Although I have a good feel for this area in general, I did not carefully check the paper's details, e.g., the math, experimental design, or novelty.

**Typos Grammar Style And Presentation Improvements:**

line 136: remove the 'and' before etc
line 523: in other words
line 631: remove deeply

---

> ### Author Rebuttal · Authors · 2023-08-28
>
> Thanks for your comments. In the submitted paper, the main contribution is to theoretically investigate the relationship between reproducibility and a splitting strategy. As a novel splitting strategy, a 3$\times$2 BCV has indeed been investigated from the perspective of statistical estimation in previous studies [1-2]. However, the previous perspective is different from our perspective of reproducibility (refer to Eq. (2) in our paper) considered in this paper. Moreover, the fourth advantage of a 3$\times$2 BCV (mentioned at the end of Section 3.1 in our paper), i.e., facilitating us to design an aggregated estimator, is firstly proposed in this paper and plays an important role in ensuring the high reproducibility of a conclusion.
>
> For your question (A), there are several other available splitting strategies in our consideration [3-4], including Monte Carlo cross-validation (CV), repeated learning-testing, K-fold CV, etc. Nevertheless, due to the limited response time and the expensive simulation-based evaluation method in our experiments, we could merely conduct a fraction of the additional experiments over the other splitting strategies. Here, considering that a 5-fold CV is widely used, we mainly conduct experiments on the reproducibility of a 5-fold CV in the NER and chunking tasks. The results are shown in the following table.
>
> |           | ST | RS |  5-fold CV  |  3x2 BCV avg  |  3x2 BCV vote  |  3x2 BCV mixture |
> |:---:|:---:|:---:|:---:|:---:|:---:|:---:|
> |   |  |  |   |  **NER**  |  |  |
>  |  Expectation ($\times$10$^{-4}$ ) | 17.33  |  17.33  |  15.63  |  5.94  |  10.50  |  11.43  |
>  |  Variance ($\times$10$^{-7}$ ) | 115.30 | 18.98 | 8.13 | 4.46 |7.93 | 6.47 |
>   |  SNR |  0.51  |  1.26  |  **1.73** | 0.89 | 1.18  |  1.42 |
>    |  Reproducibility | 0.68  |  0.89 | **0.98** | 0.83 | 0.90 |  0.94 |
> |   |  |  |   |  **Chunking**  |  |  |
>    | Expectation  ($\times$10$^{-4}$ ) | 4.82  |  4.54  |  4.20  |  11.81  |  10.12  |  12.62 |
>  | Variance ($\times$10$^{-7}$ )  | 61.19 | 8.83  | 3.47  |  1.36  |  3.10  |  1.70  |
>  |  SNR  |  0.20  |  0.48  |  0.71  | **3.20**  | 1.82  |  3.06  |
>  |  Reproducibility  |  0.59  |  0.67  |  0.70  |  **1.00**| 0.96  | **1.00** |
>
> From the above table, we obtain that our proposed mixture estimation based on a 3$\times$2 BCV owns a comparable reproducibility with a 5-fold CV in the NER task and achieves a substantially better reproducibility than a 5-fold CV in the chunking task. Specifically, on the two tasks, the three types of the 3$\times$2 BCV estimators constantly possess smaller variances than the 5-fold CV estimator. Nevertheless, in the NER task, the expectation of a 5-fold CV estimator is larger than that of 3$\times$2 BCV estimators. This observation is consistent with the assumption (Case I) mentioned in lines 272-274 in our paper. Thus, the 5-fold CV owns a slightly higher SNR and reproducibility than our proposed method. In contrast, in the chunking task, because the expectation decreases w.r.t the training set size (as mentioned in Case II in lines 275-278 in our paper), our proposed method has a higher SNR than the 5-fold CV. Thus, the proposed method achieves better reproducibility than the 5-fold CV in the chunking task. In sum, we consider the proposed method to be a better choice than a 5-fold CV.
>
> For your question (B), we further consider the m$\times$2 BCV with m=5, 7, and 9, and compare them with a 3$\times$2 BCV. Due to the limited time, we merely conducted the experiments in the NER task. The experimental results are shown in the following table.
> |  | 3$\times$2 BCV | 5$\times$2 BCV | 7$\times$2 BCV | 9$\times$2 BCV |
> |:-:|:-:|:-:|:-:|:-:|
> |  |  | **Averaged** | **estimation** |  |
>   |  Expectation  ($\times$10$^{-4}$ )  |  5.94  |  6.94  |  5.54  |  6.89  |
>  |  Variance ($\times$10$^{-7}$ )    |  4.46  |  3.07  |  2.30  |  2.05  |
>  |  SNR  |  0.89  |  1.25  |  1.16  |  **1.52**  |
>  |  Reproducibility  |  0.83  |  0.88  |  0.88  |  **0.94**  |
> |  |    | **Vote** | **estimation** |  |
>  |  Expectation  ($\times$10$^{-4}$ )  |  10.50  |  9.77  |  11.16  |  12.07  |
>  |  Variance ($\times$10$^{-7}$ )  |  7.93  |  9.68  |  9.41 |  8.51 |
>  |  SNR  |  1.18  |  0.99  |  1.15  |  **1.31**  |
>  |  Reproducibility  |  0.90  |  0.77  |  0.88  |  **0.92**  |
> |  |  | **Mixture** |  **estimation** |  |
>  |  Expectation  ($\times$10$^{-4}$ ) |  11.43  |  12.00  |  12.32  |  13.28  |
>  |  Variance ($\times$10$^{-7}$ )  |  6.47  | 6.33  |  7.16  |  6.00 |
>  |  SNR  |  1.42 | 1.51 | 1.46 | **1.71** |
>  |  Reproducibility  |  0.94  |  0.93  |  0.94  |  **0.99**|
>
> From the table, we obtain that (a) for each of the three estimators, with an increasing m, the variance of the averaged estimator decreases; (b) when m increases, the expectation of the averaged estimator remains almost unchanged, and the expectations of the vote and mixture estimation increase. In total, a larger m in m$\times$2 BCV frequently leads to a larger SNR and a higher reproducibility. Nevertheless, in contrast to a 3$\times$2 BCV, an mx2 BCV with m>3 has an expensive computational cost and a complex splitting manner, and thus it is not easy to use. Therefore, we recommend a 3$\times$2 BCV in our paper instead of a general m$\times$2 BCV.
>
> Moreover, thanks for your warm reminder about the typos and the presentation improvements. We will carefully revise the errors in the future version of our paper.
>
> [1] Wang, Y., et al. 2014. Blocked 3x2 Cross-Validated t-test for Comparing Supervised Classification Learning Algorithms. Neural Computation 26(1): 208-235.
>
> [2] Ruibo Wang and Jihong Li. 2019. Bayes Test of Precision, Recall, and F1 Measure for Comparison of Two Natural Language Processing Models. In ACL, 4135-4145.
>
> [3] Arlot, S. and A. Celisse. 2010. A survey of cross-validation procedures for model selection. Statistics surveys 4: 40-79.
>
> [4] Henry Moss, David Leslie, and Paul Rayson. 2018. Using J-K-fold Cross Validation To Reduce Variance When Tuning NLP Models. In ICCL, 2978-2989.

---

### Official Review · Reviewer_39w5 · 2023-08-13

**Soundness:** 3

**Excitement:**

4: Strong: This paper deepens the understanding of some phenomenon or lowers the barriers to an existing research direction.

**Paper Topic And Main Contributions:**

This paper addressed the reproducibility crisis in NLP model comparison, where standard splits often lead to unreliable conclusions. They proposed a new approach based on a probabilistic function and demonstrated theoretically that reproducibility is influenced by the signal-to-noise ratio (SNR) of the model performance estimator obtained from a corpus splitting strategy. Building on this, they developed a novel mixture estimator with a regularized corpus-splitting strategy and conducted experiments on multiple NLP tasks to show its effectiveness in achieving high SNR and increasing reproducibility.

**Reasons To Accept:**

1. The paper contains great motivation by exploring the reproducibility crisis in NLP.
2. The paper is well organized.
3. Experiments seem to be solid and comprehensive.
4. This paper may have contributions to the NLP community.

**Reasons To Reject:**

1. It is unclear why the Signal-to-Noise Ratio is useful in determining reproducibility from a theoretical perspective.

**Reproducibility:**

4: Could mostly reproduce the results, but there may be some variation because of sample variance or minor variations in their interpretation of the protocol or method.

**Reviewer Confidence:**

4: Quite sure. I tried to check the important points carefully. It's unlikely, though conceivable, that I missed something that should affect my ratings.

---

> ### Author Rebuttal · Authors · 2023-08-28
>
> Thanks for your comments. In our paper, we mainly consider inferential reproducibility in a comparison of two NLP models. Inferential reproducibility is about the conclusions drawn, and it indicates to what extent a conclusion is reproducible if one can draw it from a different experimental setup [1].
>
> Many empirical studies [2-5] have achieved a consensus that the variability in the empirical results is critical to the reproducibility of a model comparison besides the reported numerical results of the performance of two NLP models, and the former is even more important than the latter. Qualitatively speaking, smaller variability and a large difference between the observed results of the performance of two NLP models lead to a highly reproducible conclusion.
>
> From a statistical perspective, our paper illustrates that the signal-to-noise ratio (SNR), which combines the above-mentioned variability and the expected difference, can be used to characterize a theoretical lower bound of the reproducibility (refer to Eq. (3) in our paper). A larger SNR leads to higher reproducibility. When the SNR tends to infinity, the reproducibility tends to one, correspondingly.
>
> Specifically, the performance of two NLP models can be regarded as two random variables, namely, $\hat{\nu}_A$ and $\hat{\nu}_B$, attributing to the randomness in the NLP modeling process. The reported numerical results in an empirical study are considered as the observations of the two random variables. Furthermore, the conclusion in a comparison of two NLP models can be introduced as $\nu_A > \nu_B$ where $\nu_A = E[\hat{\nu}_A]$ and $\nu_B = E[\hat{\nu}_B]$. In other words, the expected performance of model $A$ is superior to that of model $B$. On the basis of these notations, we further define reproducibility as the following conditional probability.
> $$P(\hat{\nu}_A > \hat{\nu}_B  | \nu_A > \nu_B).  \tag{1}$$
>
> The reproducibility in Eq. (1) is interpreted as the probability of reproducing the conclusion $\nu_A > \nu_B$ from the observed results $\hat{\nu}_A$ and $\hat{\nu}_B$. By defining a difference of $\hat{\nu}_d = \hat{\nu}_A-\hat{\nu}_B$, we rewrite the reproducibility as the following form.
> $$P(\hat{\nu}_d > 0 | \nu_A > \nu_B). \tag{2}$$
>
> From Eq. (2), we obtain that the reproducibility heavily depends on the shape of the conditional density $P(\hat{\nu}_d| \nu_A > \nu_B)$. If the shape is very thin and far from zero, indicating the variability of the reported results is small and their difference is large, the reproducibility is high. Because the location and scale of the density can be characterized by the expectation and the variance of $\hat{\nu}_d$, the shape of the density can be characterized by the SNR of $\hat{\nu}_d$. According to the one-side Chebyshev inequality, we can obtain a theoretical lower bound of the reproducibility as follows.
> $$P(\hat{\nu}_d > 0 | \nu_A > \nu_B)\ge\frac{SNR^2}{1+SNR^2}  \tag{3}$$
> where the SNR is the signal-to-noise ratio of $\hat{\nu}_d$ (refer to Eq. (4) in our paper). From Eq. (3), when the variance of $\hat{\nu}_d$ decreases and the expectation of $\hat{\nu}_d$ is slightly large, then the SNR of $\hat{\nu}_d$ becomes large, and the reproducibility would correspondingly become high. This theoretical interpretation is consistent with the empirical observations in [2-5].
>
> In sum, the signal-to-noise ratio is very useful in determining reproducibility and facilitates us to find a better estimation method of model performance to produce a highly reproducible conclusion in a comparison of NLP models.
>
> [1] Xavier Bouthillier, C$\acute{e}$sar Laurent, and Pascal Vincent. 2019. Unreproducible Research is Reproducible. In ICML, pages 725-734.
>
> [2] Anya Belz, Shubham Agarwal, Anastasia Shimorina, and Ehud Reiter. 2021. A Systematic Review of Reproducibility Research in Natural Language Processing. In EACL, pages 381-393.
>
> [3] Taylor Berg-Kirkpatrick, David Burkett, and Klein Dan. 2012. An Empirical Investigation of Statistical Significance in nlp. In EMNLP, pages 995-1005.
>
> [4] Henry Moss, David Leslie, and Paul Rayson. 2018. Using J-K-fold Cross Validation To Reduce Variance When Tuning NLP Models. In ICCL, pages 2978-2989.
>
> [5] Kyle Gorman and Steven Bedrick. 2019. We Need to Talk about Standard Splits. In ACL, pages 2786-2791.

---

### Meta-Review · Area_Chair_K19v · 2023-09-17

**Recommendation:** 5

**Metareview:**

The paper addresses the impact of splitting strategies on reproducibility. It provides theoretical motivations for using blocked 3x2 cross-validation and presents experiments showing that this indeed provides a higher signal-to-noise ratio. The paper is presented nicely with theoretical motivations that are backed up by results.

There were concerns about the appropriateness of the signal-to-noise ratio as a measure for reproducibility and that the evaluation is optimised to show a specific result. The rebuttal provides extensive explanations of why signal-to-noise is suitable and convincingly explains why the experimental setup is different from the regular case where we want to test whether something is it case.
It would be good if a future version provides these explanations (in main text of appendix, depending on space) so that readers understand it at first hand.

---

### Decision · Program_Chairs · 2023-10-07

**Decision:**

Accept-Main

**Comment:**

The paper addresses the impact of splitting strategies on reproducibility. It provides theoretical motivations for using blocked 3x2 cross-validation and presents experiments showing that this indeed provides a higher signal-to-noise ratio. The paper is presented nicely with theoretical motivations that are backed up by results.

There were concerns about the appropriateness of the signal-to-noise ratio as a measure for reproducibility and that the evaluation is optimised to show a specific result. The rebuttal provides extensive explanations of why signal-to-noise is suitable and convincingly explains why the experimental setup is different from the regular case where we want to test whether something is it case.
It would be good if a future version provides these explanations (in main text of appendix, depending on space) so that readers understand it at first hand.